# The Adversarial Consistency of Surrogate Risks for Binary Classification

**Natalie S. Frank**
Courant Institute
New York University
New York, NY 10012
nf1066@nyu.edu

**Jonathan Niles-Weed**
Courant Institute
New York University
New York, NY 10012
jnw@cims.nyu.edu

## Abstract

We study the consistency of surrogate risks for robust binary classification. It is common to learn robust classifiers by adversarial training, which seeks to minimize the expected 0-1 loss when each example can be maliciously corrupted within a small ball. We give a simple and complete characterization of the set of surrogate loss functions that are *consistent*, i.e., that can replace the 0-1 loss without affecting the minimizing sequences of the original adversarial risk, for any data distribution. We also prove a quantitative version of adversarial consistency for the $\rho$-margin loss. Our results reveal that the class of adversarially consistent surrogates is substantially smaller than in the standard setting, where many common surrogates are known to be consistent.

## 1 Introduction

A central issue in the study of neural nets is their susceptibility to adversarial perturbations—perturbations imperceptible to the human eye can cause a neural net to misclassify an image [Szegedy et al., 2013, Biggio et al., 2013]. The same phenomenon appears in other types of data such as speech and text. As deep nets are used in applications such as self-driving cars and medical imaging [Paschali et al., 2018, Li et al., 2021], training classifiers robust to adversarial perturbations is a central question in machine learning.

The foundational theory of surrogates for classication in well understood. In the standard classification setting, one seeks to minimize the *classification* risk— the proportion of incorrectly classified data. Since minimizing the classification risk is typically computationally intractable [Ben-David et al., 2003], a common approach is to instead minimize a better-behaved alternative called the *surrogate risk*. However, one must verify that classifiers with low surrogate risk also achieve low classification risk. If for every data distribution, a sequence of functions minimizing the surrogate also minimizes the classification risk, the surrogate risk is called *consistent*. Many classic papers study the consistency of surrogate risks in the standard classification setting [Bartlett et al., 2006, Lin, 2004, Steinwart, 2007, Philip M. Long, 2013, Mingyuan Zhang, 2020].

Unlike the standard case, however, little is known about the consistency of surrogate risks in the context of adversarial training, which involves risks that compute the supremum of a surrogate loss function over an $\epsilon$-ball. Though this question has been partially studied in the literature [Awasthi et al., 2021a,c, Meunier et al., 2022], a general theory is lacking. Existing results reveal, however, that the situation is substantially different from the standard case: for instance, [Meunier et al., 2022] show that no *convex* surrogate can be adversarially consistent. To our knowledge, no adversarially consistent risks are known.

In this work, we give a complete characterization of adversarial consistency for surrogate losses.

37th Conference on Neural Information Processing Systems (NeurIPS 2023).

**Our Contributions:**

- In Section 4 we give a surprisingly simple necessary and sufficient condition for adversarial consistency:

  **Informal Theorem.** *Under reasonable assumptions on the surrogate loss $\phi$, the supremum-based $\phi$-risk is adversarially consistent if and only if $\inf_\alpha \phi(\alpha)/2 + \phi(-\alpha)/2 < \phi(0)$.*

  In particular, this result proves consistency for any loss function that is *not* midpoint convex at the origin.

- In Section 5, we specialize to the case of the $\rho$-margin loss, where we obtain a quantitative proof of adversarial consistency by explicitly bounding the excess adversarial risk.

To the best of the authors' knowledge, this paper is the first to prove that a loss-based learning procedure is consistent for a wide range of distributions in the adversarial setting. As mentioned above, the $\rho$-margin loss $\phi_\rho(\alpha) = \min(1, \max(1 - \alpha/\rho, 0))$ satisfies the conditions of Informal Theorem above, as does the shifted sigmoid loss $\phi_\tau(\alpha) = 1/(1 + \exp(\alpha - \tau))$ with $\tau > 0$, which confirms a conjecture of Meunier et al. [2022]. By contrast, all convex losses satisfy $\inf_\alpha \phi(\alpha)/2 + \phi(-\alpha)/2 = \phi(0)$, and are therefore not adversarially consistent.

In addition to consistency, one would hope to obtain a quantitative comparison between the adversarial surrogate risk and the adversarial classification risk. Our bound in Section 5 shows that the excess error of the adversarial $\rho$-margin loss is a linear upper bound on the adversarial classification error, which implies that minimizing the adversarial $\rho$-margin loss is an effective procedure for minimizing the adversarial classification error. Extending the bound in Section 5 to further losses remains an open question.

## 2   Related Works

Many previous works have studied the consistency of surrogate risks [Bartlett et al., 2006, Lin, 2004, Steinwart, 2007, Philip M. Long, 2013, Mingyuan Zhang, 2020]. The classic papers by [Bartlett et al., 2006, Lin, 2004, Zhang, 2004] explore the consistency of surrogate risks over all measurable functions. The works [Philip M. Long, 2013, Mingyuan Zhang, 2020, Awasthi et al., 2022] study $\mathcal{H}$-consistency, which is consistency restricted to a smaller set of functions. Steinwart [2007] generalizes some of these results into a framework referred to as *calibration*. Awasthi et al. [2021a], Bao et al. [2021], Awasthi et al. [2021c], Meunier et al. [2022] then use this framework to analyze the calibration of adversarial surrogate losses. Furthermore Meunier et al. [2022] relate calibration to consistency for adversarial losses in certain cases — they show that no convex loss is adversarially consistent. They also conjecture that a class of surrogate losses called the *odd shifted* losses are adversarially consistent. Meunier et al. [2022] also show that in a restricted setting, surrogates are consistent for 'optimal attacks'. The proof of our result formalizes this intuition. Simultaneous work [Mao et al., 2023] shows that the $\rho$-margin loss is adversarially $\mathcal{H}$-consistent for typical function classes. Lastly, Bhattacharjee and Chaudhuri [2020, 2021] use a different set of techniques to study the consistency of non-parametric methods in adversarial scenarios.

Our results rely on recent works establishing the properties of minimizers to surrogate adversarial risks. [Awasthi et al., 2021b, Pydi and Jog, 2021, Bungert et al., 2021] all proved the existence of minimizers to the adversarial risk and [Pydi and Jog, 2021] proved a minimax theorem for the zero-one loss. Building on the work of [Pydi and Jog, 2021], [Frank and Niles-Weed, 2023] later proved similar existence and minimax statements for arbitrary surrogate losses. Trillos et al. [2022, 2023] extend some of these results to the multiclass case. Lastly, [Trillos and Murray, 2020] study further properties of the minimizers to the adversarial classification loss.

## 3   Problem Setup

This section contains the necessary background for our results. Section 3.1 gives precise definitions for the main concepts, and Section 3.2 describes the minimax theorems that are at the heart of our proof.

## 3.1 Surrogate Risks

This paper studies binary classification on $\mathbb{R}^d$. Explicitly, labels are $\{-1, +1\}$ and the data is distributed according to a distribution $\mathcal{D}$ on the set $\mathbb{R}^d \times \{-1, +1\}$. The measures $\mathbb{P}_1$, $\mathbb{P}_0$ define the relative probabilities of finding points with a given label in a region of $\mathbb{R}^d$. Formally, define measures on $\mathbb{R}^d$ by

$$\mathbb{P}_1(A) = \mathcal{D}(A \times \{+1\}), \mathbb{P}_0(A) = \mathcal{D}(A \times \{-1\}).$$

The *classification risk* $R(f)$ is then the probability of misclassifying a point under $\mathcal{D}$:

$$R(f) = \int \mathbf{1}_{f(\mathbf{x}) \leq 0} d\mathbb{P}_1 + \int \mathbf{1}_{f(\mathbf{x}) > 0} d\mathbb{P}_0. \tag{1}$$

The surrogate to $R$ is

$$R_\phi(f) = \int \phi(f) d\mathbb{P}_1 + \int \phi(-f) d\mathbb{P}_0. \tag{2}$$

A classifier can be obtained by minimizing either $R$ or $R_\phi$ over the set of all measurable functions. A point $\mathbf{x}$ is then classified according to $\text{sign } f$. There are many possible choices for $\phi$—typically one chooses a loss that is easy to optimize. In this paper, we assume that

**Assumption 1.** *$\phi$ is non-increasing, non-negative, continuous, and $\lim_{\alpha \to \infty} \phi(\alpha) = 0$.*

Most surrogate losses in machine learning satisfy this assumption. Learning algorithms typically optimize the risk in (2) using an iterative procedure, which produces a sequence of functions that minimizes $R_\phi$. We call $R_\phi$ a *consistent risk* and $\phi$ a *consistent loss* if for all distributions, every minimizing sequence of $R_\phi$ is also a minimizing sequence of $R$.[1] Alternatively, the risks $R$, $R_\phi$ can be expressed in terms of the quantities $\mathbb{P} = \mathbb{P}_0 + \mathbb{P}_1$ and $\eta = d\mathbb{P}_1/d\mathbb{P}$. For all $\eta \in [0, 1]$, define

$$C(\eta, \alpha) = \eta \mathbf{1}_{\alpha \leq 0} + (1 - \eta)\mathbf{1}_{\alpha > 0}, \quad C^*(\eta) = \inf_\alpha C(\eta, \alpha), \tag{3}$$

$$C_\phi(\eta, \alpha) = \eta \phi(\alpha) + (1 - \eta)\phi(-\alpha), \quad C_\phi^*(\eta) = \inf_\alpha C_\phi(\eta, \alpha) \tag{4}$$

For more on the definitions of $R, R_\phi, C, C_\phi$, see [Bartlett et al., 2006] or Sections 3.1 and 3.2 of [Frank and Niles-Weed, 2023]. Using these definitions, $R(f) = \int C(\eta(\mathbf{x}), f(\mathbf{x}))d\mathbb{P}$ and

$$R_\phi(f) = \int C_\phi(\eta(\mathbf{x}), f(\mathbf{x}))d\mathbb{P} \tag{5}$$

This alternative view of the risks $R$ and $R_\phi$ provides a 'pointwise' criterion for consistency— if the function $f(\mathbf{x})$ minimizes $C_\phi(\eta(\mathbf{x}), \cdot)$ at each point, then it also minimizes $R_\phi$. However, minimizers to $C_\phi(\eta, \cdot)$ over $\mathbb{R}$ do not always exist— consider for instance $\eta = 1$ for the exponential loss $\phi(\alpha) = e^{-\alpha}$. In general, for minimizers of $C_\phi(\eta, \cdot)$ to exist, one must work over the extended real numbers $\overline{\mathbb{R}} = \mathbb{R} \cup \{-\infty, +\infty\}$. The following proposition proved in Appendix A implies that 'pointwise' considerations also extends to minimizing sequences of functions.

**Proposition 1.** *The following are equivalent:*

*1) $\phi$ is consistent*

*2) Every minimizing sequence of $C_\phi(\eta, \cdot)$ is also a minimizing sequence of $C(\eta, \cdot)$*

*3) Every $\overline{\mathbb{R}}$-valued minimizer of $R_\phi$ is a minimizer of $R$*

This result is well-known in prior literature; in particular the equivalence between 2) and 3) is closely related to the equivalence between calibration and consistency in the non-adversarial setting [Steinwart, 2007]. Most importantly, the equivalence between 1) and 3) reduces studying minimizing sequences of functionals to studying minimizers of functions. We will show that the equivalence

---

[1] In the context of standard (non-adversarial) learning, the concept we defined as consistency is often referred to as *calibration*, see for instance [Bartlett et al., 2006, Steinwart, 2007]. We opt for the term 'consistency' as the prior works [Awasthi et al., 2021a,c, Meunier et al., 2022] use calibration to refer to a different but related concept in the adversarial setting.

between 1) and 2) has an analog in the adversarial scenario, but the equivalence between 1) and 3) does not.

In the adversarial classification setting, every $x$-value is perturbed by a malicious adversary before undergoing classification by $f$. We assume that these perturbations are bounded by $\epsilon$ in some norm $\|\cdot\|$ and furthermore, the adversary knows both our classifier $f$ and the true label of the point $\mathbf{x}$. In other words, $f$ misclassifies $(\mathbf{x}, y)$ when there is a point $\mathbf{x}' \in \overline{B_\epsilon(\mathbf{x})}$ for which $\mathbf{1}_{f(\mathbf{x}')\leq 0} = 1$ for $y = +1$ and $\mathbf{1}_{f(\mathbf{x}')>0} = 1$ for $y = -1$. Conveniently, this criterion can be expressed in terms of suprema. For any function $g$, we define

$$S_\epsilon(g)(\mathbf{x}) = \sup_{\|\mathbf{h}\|\leq\epsilon} g(\mathbf{x} + \mathbf{h})$$

A point $\mathbf{x}$ with label $+1$ is misclassified when $S_\epsilon(\mathbf{1}_{f\leq0})(\mathbf{x}) = 1$ and a point $\mathbf{x}$ with label $-1$ is misclassified when $S_\epsilon(\mathbf{1}_{f>0})(\mathbf{x}) = 1$. Hence the expected fraction of errors under the adversarial attack is

$$R^\epsilon(f) = \int S_\epsilon(\mathbf{1}_{f\leq0})d\mathbb{P}_1 + \int S_\epsilon(\mathbf{1}_{f>0})d\mathbb{P}_0, \tag{6}$$

which is called the *adversarial classification risk* [2]. Again, optimizing the empirical version of (6) is computationally intractable so instead one minimizes a surrogate of the form

$$R^\epsilon_\phi(f) = \int S_\epsilon(\phi \circ f)d\mathbb{P}_1 + \int S_\epsilon(\phi \circ -f)d\mathbb{P}_0 \tag{7}$$

Due to the supremum in this expression, we refer to such a risk as a *supremum-based surrogate*. We define adversarial consistency as

**Definition 1.** *The risk $R^\epsilon_\phi$ is adversarially consistent if for every data distribution, every sequence $f_n$ which minimizes $R^\epsilon_\phi$ over all Borel measurable functions also minimizes $R^\epsilon$. We say that the loss $\phi$ is adversarially consistent if the risk $R^\epsilon_\phi$ is adversarially consistent.*

Many convex and non-convex losses are consistent in standard classification [Bartlett et al., 2006, Zhang, 2004, Steinwart, 2007, Reid and Williamson, 2009, Lin, 2004]. By contrast, adversarial consistency often fails. For instance, Meunier et al. [2022] show that convex losses are not adversarially consistent. Furthermore, their example shows that the equivalence between 1) and 3) in Proposition 1 does *not* hold in the adversarial context. Thus, to understand adversarial consistency, it does not suffice to compare minimizers of $R^\epsilon_\phi$ and $R^\epsilon$. To illustrate this distinction, we show the following result, adapted from [Meunier et al., 2022].

**Proposition 2.** *Assume that $\inf_\alpha \phi(\alpha)/2 + \phi(-\alpha)/2 = \phi(0)$. Then $\phi$ is not adversarially consistent.*

*Proof.* Let $\mathbb{P}_0 = \mathbb{P}_1$ be the the uniform distribution on the ball $\overline{B_R(\mathbf{0})}$ and let $\epsilon = 2R$. Let $\phi$ be a loss function for which $\inf_\alpha \phi(\alpha)/2 + \phi(-\alpha)/2 = C^*_\phi(1/2) = \phi(0)$. Notice that $\inf_f R^\epsilon(f) \geq \inf_f R(f)$ and $\inf_f R^\epsilon_\phi(f) \geq \inf_f R_\phi(f)$. Since $\mathbb{P}_0 = \mathbb{P}_1$, the optimal non-adversarial risk is $\inf_f R(f) = 1/2$. Moreover, as $C^*_\phi(1/2) = \phi(0)$, the optimal non-adversarial surrogate risk is $\inf_f R_\phi(f) = C^*_\phi(1/2) = \phi(0)$. Thus, for the function $f^* \equiv 0$, $R^\epsilon(f^*) = \inf_f R(f) = 1/2$ and $R^\epsilon_\phi(f^*) = \inf_f R_\phi(f) = \phi(0)$. Therefore $f^*$ minimizes both $R^\epsilon_\phi$ and $R^\epsilon$. Now consider the sequence of functions

$$f_n(\mathbf{x}) = \begin{cases} \frac{1}{n} & \mathbf{x} = 0 \\ -\frac{1}{n} & \mathbf{x} \neq 0 \end{cases}$$

Because $\epsilon = 2R$, every point in the support of the distribution can be perturbed to every other point. Thus $S_\epsilon(\phi \circ f_n)(\mathbf{x}) = \phi(-1/n)$ and $S_\epsilon(\phi \circ -f_n)(\mathbf{x}) = \phi(-1/n)$. However, $S_\epsilon(\mathbf{1}_{f\leq0}) = 1$ and $S_\epsilon(\mathbf{1}_{f>0}) = 1$. Therefore, $R^\epsilon_\phi(f_n) = \phi(-1/n)$ while $R^\epsilon(f_n) = 1$ for all $n$. As $\phi$ is continuous, $\lim_{n\to\infty} R^\epsilon_\phi(f_n) = \phi(0)$. Thus $f_n$ is a minimizing sequence of $R^\epsilon_\phi$ but not of $R^\epsilon$, so $\phi$ is not adversarially consistent. $\qquad\square$

---

[2] Defining this integral requires some care because for a Borel function $g$, $S_\epsilon(g)$ may not be measurable; see Section 3.3 and Appendix A of [Frank and Niles-Weed, 2023] for details.

This example shows that if $C_\phi^*(1/2) = \phi(0)$, then $\phi$ is not adversarially consistent. The main result of this paper is that this is the *only* obstruction to adversarial consistency: $\phi$ is adversarially consistent if and only if $C_\phi^*(1/2) < \phi(0)$.

We begin by showing that this condition suffices for consistency in the *non-adversarial* setting. Surprisingly, despite the wealth of work on this topic, this condition does not appear to be known.

**Proposition 3.** *If $C_\phi^*(1/2) < \phi(0)$, then $\phi$ is consistent.*

See Appendix C for a proof.

Again, some losses that satisfy this property are the $\rho$-margin loss $\phi_\rho(\alpha) = \min(1, \max(1 - \alpha/\rho, 0))$ and the the shifted sigmoid loss proposed by Meunier et al. [2022], $\phi(\alpha) = 1/(1 + \exp(\alpha - \tau))$, $\tau > 0$. (In fact, one can show that the class of shifted odd losses proposed by Meunier et al. [2022] satisfy $C_\phi^*(1/2) < \phi(0)$.)

Notice that all convex losses satisfy $C_\phi^*(1/2) = \phi(0)$:

$$C_\phi^*(1/2) = \inf_\alpha \frac{1}{2}\phi(\alpha) + \frac{1}{2}\phi(-\alpha) \geq \phi(0)$$

The opposite inequality follows from the observation that $C_\phi^*(1/2) \leq C_\phi(1/2, 0) = \phi(0)$. In contrast, recall that a convex loss $\phi$ with $\phi'(0) < 0$ is consistent [Bartlett et al., 2006].

As conjectured by prior work Bao et al. [2021], Meunier et al. [2022], the fundamental reason losses with $C_\phi^*(1/2) < \phi(0)$ are adversarially consistent is that minimizers of $C_\phi(\eta, \cdot)$ are uniformly bounded away from 0 for all $\eta$:

**Lemma 1.** *The loss $\phi$ satisfies $C_\phi^*(1/2) < \phi(0)$ iff there is an $a > 0$ for which any minimizer $\alpha^*$ of $C_\phi(\eta, \cdot)$ satisfies $|\alpha| \geq a$.*

See C for a proof. Concretely, one can show that for the $\rho$-margin loss $\phi_\rho$, a minimizer $\alpha^*$ of $C_{\phi_\rho}(\eta, \cdot)$ must satisfy $|\alpha^*| \geq \rho$. Similarly, a minimizer $\alpha^*$ of $C_{\phi_\tau}(\eta, \cdot)$ of the shifted sigmoid loss $\phi_\tau = 1/(1 + \exp(\alpha - \tau))$, $\tau > 0$ is always either $-\infty$ or $+\infty$. In 4, we use this property to show that minimizing sequences of $R_\phi^\epsilon$ must be uniformly bounded away from zero, thus ruling out the counterexample presented in Proposition 2.

## 3.2 Minimax Theorems for Adversarial Risks

We study the consistency of $\phi$ by by comparing minimizing sequences of $R_\phi^\epsilon$ with those of $R^\epsilon$. In the next section, in order to compare these minimizing sequences, we will attempt to re-write the adversarial loss in a 'pointwise' manner similar to Proposition 1. In order to achieve this representation of the adversarial loss, we apply minimax and complimentary slackness theorems from [Pydi and Jog, 2021, Frank and Niles-Weed, 2023].

Before presenting these results, we introduce the $\infty$-Wasserstein metric from optimal transport. For two finite probability measures $\mathbb{Q}, \mathbb{Q}'$ satisfying $\mathbb{Q}(\mathbb{R}^d) = \mathbb{Q}'(\mathbb{R}^d)$, let $\Pi(\mathbb{Q}, \mathbb{Q}')$ be the set of *couplings* between $\mathbb{Q}$ and $\mathbb{Q}'$:

$$\Pi(\mathbb{Q}, \mathbb{Q}') = \{\gamma : \text{measure on } \mathbb{R}^d \times \mathbb{R}^d \text{ with } \gamma(A \times \mathbb{R}^d) = \mathbb{Q}(A), \gamma(\mathbb{R}^d \times A) = \mathbb{Q}'(A)\}$$

The distance between $\mathbb{Q}'$ and $\mathbb{Q}$ in the Wasserstein $\infty$-metric $W_\infty$ is defined as

$$W_\infty(\mathbb{Q}, \mathbb{Q}') = \inf_{\gamma \in \Pi(\mathbb{Q}, \mathbb{Q}')} \operatorname*{ess\,sup}_{(\mathbf{x}, \mathbf{y}) \sim \gamma} \|\mathbf{x} - \mathbf{y}\|.$$

The $W_\infty$ distance is in fact a metric on the space of measures. We denote the $\infty$-Wasserstein ball around a measure $\mathbb{Q}$ by

$$\mathcal{B}_\epsilon^\infty(\mathbb{Q}) = \{\mathbb{Q}' : \mathbb{Q}' \text{ Borel}, \quad W_\infty(\mathbb{Q}, \mathbb{Q}') \leq \epsilon\}$$

Informally, the measure $\mathbb{Q}'$ is in $\mathcal{B}_\epsilon^\infty(\mathbb{Q})$ if perturbing points by at most $\epsilon$ under the measure $\mathbb{Q}$ can produce $\mathbb{Q}'$. As a result, Wasserstein $\infty$-balls are fairly useful for modeling adversarial attacks. Specifically, one can show:

**Lemma 2.** *For any function $g$ and measures $\mathbb{Q}', \mathbb{Q}$ with $W_\infty(\mathbb{Q}', \mathbb{Q}) \leq \epsilon$, the inequality $\int S_\epsilon(g)d\mathbb{Q} \geq \int gd\mathbb{Q}'$ holds.*

See Appendix D for a proof.

Minimax theorems from prior work use this framework to introduce dual problems to the adversarial classification risks (6) and (7). Let $\mathbb{P}'_0, \mathbb{P}'_1$ be finite Borel measures and define

$$\bar{R}(\mathbb{P}'_0, \mathbb{P}'_1) = \int C^* \left( \frac{d\mathbb{P}'_1}{d(\mathbb{P}'_0 + \mathbb{P}'_1)} \right) d(\mathbb{P}'_0 + \mathbb{P}'_1) \tag{8}$$

where $C^*$ is defined by (3). The next theorem states that maximizing $\bar{R}$ over $W_\infty$ balls is in fact a dual problem to minimizing $R^\epsilon$.

**Theorem 1.** *Let $\bar{R}$ be defined by (8).*

$$\inf_{\substack{f \text{ Borel} \\ \mathbb{R}\text{-valued}}} R^\epsilon(f) = \sup_{\substack{\mathbb{P}'_0 \in \mathcal{B}^\infty_\epsilon(\mathbb{P}_0) \\ \mathbb{P}'_1 \in \mathcal{B}^\infty_\epsilon(\mathbb{P}_1)}} \bar{R}(\mathbb{P}'_0, \mathbb{P}'_1) \tag{9}$$

*and furthermore equality is attained for some Borel measurable $\hat{f}$ and $\hat{\mathbb{P}}_1, \hat{\mathbb{P}}_0$ with $W_\infty(\hat{\mathbb{P}}_0, \mathbb{P}_0) \leq \epsilon$ and $W_\infty(\hat{\mathbb{P}}_1, \mathbb{P}_1) \leq \epsilon$.*

The first to show such a theorem was Pydi and Jog [2021]. In comparison to their Theorem 8, Theorem 1 removes the assumption that $\mathbb{P}_0, \mathbb{P}_1$ are absolutely continuous with respect to Lebesgue measure and shows that the minimizer $\hat{f}$ is in fact Borel. We prove this theorem in Appendix E. Frank and Niles-Weed [2023] prove a similar statement for the surrogate risk $R^\epsilon_\phi$. This time, the dual objective is

$$\bar{R}_\phi(\mathbb{P}'_0, \mathbb{P}'_1) = \int C^*_\phi \left( \frac{d\mathbb{P}'_1}{d(\mathbb{P}'_0 + \mathbb{P}'_1)} \right) d(\mathbb{P}'_0 + \mathbb{P}'_1) \tag{10}$$

with $C^*_\phi$ defined by (4).

**Theorem 2.** *Assume that Assumption 1 holds, and define $\bar{R}_\phi$ by (10). Then*

$$\inf_{\substack{f \text{ Borel,} \\ f \mathbb{R}\text{-valued}}} R^\epsilon_\phi(f) = \sup_{\substack{\mathbb{P}'_0 \in \mathcal{B}^\infty_\epsilon(\mathbb{P}_0) \\ \mathbb{P}'_1 \in \mathcal{B}^\infty_\epsilon(\mathbb{P}_1)}} \bar{R}_\phi(\mathbb{P}'_0, \mathbb{P}'_1) \tag{11}$$

*and furthermore equality in the dual problem is attained for some $\mathbb{P}^*_1, \mathbb{P}^*_0$ with $W_\infty(\mathbb{P}^*_0, \mathbb{P}_0) \leq \epsilon$ and $W_\infty(\mathbb{P}^*_1, \mathbb{P}_1) \leq \epsilon$.*

Frank and Niles-Weed [2023] proved this statement in Theorem 6 but with the infimum taken over $\overline{\mathbb{R}}$-valued functions. To extend the result to $\mathbb{R}$-valued functions as in Theorem 2, we show that $\inf_{f \text{ Borel}, f \overline{\mathbb{R}}\text{-valued}} R^\epsilon_\phi(f) = \inf_{f \text{ Borel}, f \mathbb{R}\text{-valued}} R^\epsilon_\phi(f)$ in Appendix B.

## 4 Adversarially Consistent Losses

This section contains our main results on adversarial consistency. In light of Proposition 2, our main task is to show that a loss satisfying $C^*_\phi(1/2) < \phi(0)$ is adversarially consistent.

At a high level, we will show that every minimizing sequence of $R^\epsilon_\phi$ must also minimize $R^\epsilon$. However, directly analyzing minimizing sequences $\{f_n\}$ of $R^\epsilon_\phi$ and $R^\epsilon$ is challenging due to the supremums in the definitions of the adversarial risks. We therefore develop alternate characterizations of minimizing sequences to both functionals, based on complimentary slackness conditions derived from the convex duality results of Section 3.2. However, unlike standard complementary slackness conditions well known from convex optimization, these theorems allow us to characterize minimizing sequences as well as minimizers.

### 4.1 Approximate Complimentary Slackness

We first state this slackness result for the surrogate case, due to Frank and Niles-Weed [2023, Lemmas 16 and 26] and Theorem 2.

**Proposition 4.** *Let* $(\mathbb{P}_0^*, \mathbb{P}_1^*)$ *be any maximizers of* $\bar{R}_\phi$ *over* $\mathcal{B}_\epsilon^\infty(\mathbb{P}_i)$. *Define* $\mathbb{P}^* = \mathbb{P}_0^* + \mathbb{P}_1^*$, $\eta^* = d\mathbb{P}_1^*/d\mathbb{P}^*$. *If* $f_n$ *is a minimizing sequence for* $R_\phi^\epsilon$, *then the following hold:*

$$\lim_{n\to\infty} \int C_\phi(\eta^*, f_n) d\mathbb{P}^* = \int C_\phi^*(\eta^*) d\mathbb{P}^*. \tag{12}$$

$$\lim_{n\to\infty} \int S_\epsilon(\phi \circ f_n) d\mathbb{P}_1 - \int \phi \circ f_n d\mathbb{P}_1^* = 0, \quad \lim_{n\to\infty} \int S_\epsilon(\phi \circ -f_n) d\mathbb{P}_0 - \int \phi \circ -f_n d\mathbb{P}_0^* = 0 \tag{13}$$

*Proof.* Let $R_{\phi,*}^\epsilon$ be the minimal value of $R_\phi^\epsilon$ and choose a $\delta > 0$. Then for sufficiently large $N$, $n \geq N$ implies that $R_\phi^\epsilon(f_n) \leq R_{\phi,*}^\epsilon + \delta$. Lemma 2 and the definition of $C_\phi^*$ in (4) further imply that

$$R_{\phi,*}^\epsilon + \delta \geq \int S_\epsilon(\phi \circ f_n) d\mathbb{P}_1 + \int S_\epsilon(\phi \circ -f_n) d\mathbb{P}_0 \geq \int \phi \circ f_n d\mathbb{P}_1^* + \int \phi \circ -f_n d\mathbb{P}_0^* \geq R_{\phi,*}^\epsilon \tag{14}$$

As $R_{\phi,*}^\epsilon = \int C_\phi^*(\eta^*) d\mathbb{P}^*$, this relation immediately implies (12).

Next, Lemma 2 again implies that

$$\int S_\epsilon(\phi \circ f_n) d\mathbb{P}_1 \geq \int \phi \circ f_n d\mathbb{P}_1^* \quad \text{and} \quad \int S_\epsilon(\phi \circ -f_n) d\mathbb{P}_0 \geq \int \phi \circ -f_n d\mathbb{P}_0^* \tag{15}$$

while (14) implies that

$$R_{\phi,*}^\epsilon - \int \phi \circ f_n d\mathbb{P}_1^* + \int \phi \circ -f_n d\mathbb{P}_0^* \leq 0.$$

Therefore, subtracting $\int \phi \circ f_n d\mathbb{P}_1^* + \int \phi \circ -f_n d\mathbb{P}_0^*$ from (14) results in

$$\delta \geq \left( \int S_\epsilon(\phi \circ f_n) d\mathbb{P}_1 - \int \phi \circ f_n d\mathbb{P}_1^* \right) + \left( \int S_\epsilon(\phi \circ -f_n) d\mathbb{P}_0 - \int \phi \circ -f_n d\mathbb{P}_0^* \right) \geq 0. \tag{16}$$

Again, (15) implies that the quantities on parentheses are both positive which implies (13).

$\square$

Proposition 4 shows that minimizing sequences of $R_\phi^\epsilon$ satisfy two properties: 1) The sequence $\{f_n\}$ must minimize the *standard* $\phi$-risk $R_\phi$ with measures $\mathbb{P}_0^*$, $\mathbb{P}_1^*$ in place of $\mathbb{P}_0, \mathbb{P}_1$, 2) At the limit, the measures $\mathbb{P}_0^*, \mathbb{P}_1^*$ are best adversarial attacks on $\phi \circ f_n, \phi \circ -f_n$. In fact, one can show that $\{f_n\}$ is a minimizing sequence of $R_\phi^\epsilon$ *if and only if* it satisfies these properties. Crucially, a very similar characterization holds for minimizers of the adversarial classification loss. We state and prove the 'only if' direction of this characterization in Proposition 5.

**Proposition 5.** *Let* $f_n$ *be a sequence and let* $\mathbb{P}_0^*$, $\mathbb{P}_1^*$ *be measures in* $\mathcal{B}_\epsilon^\infty(\mathbb{P}_i)$. *Define* $\mathbb{P}^* = \mathbb{P}_0^* + \mathbb{P}_1^*$, $\eta^* = d\mathbb{P}_1^*/d\mathbb{P}^*$. *If the following two conditions hold:*

$$\lim_{n\to\infty} \int C(\eta^*, f_n) d\mathbb{P}^* = \int C^*(\eta^*) d\mathbb{P}^* \tag{17}$$

$$\lim_{n\to\infty} \int S_\epsilon(\mathbf{1}_{f_n \leq 0}) d\mathbb{P}_1 - \int \mathbf{1}_{f_n \leq 0} d\mathbb{P}_1^* = 0, \quad \lim_{n\to\infty} \int S_\epsilon(\mathbf{1}_{f_n > 0}) d\mathbb{P}_0 - \int \mathbf{1}_{f_n > 0} d\mathbb{P}_0^* = 0, \tag{18}$$

*then* $f_n$ *is a minimizing sequence of* $R^\epsilon$.

*Proof.* Equation 17 implies that the limit $\lim_{n\to\infty} C(\eta^*, f_n) d\mathbb{P}^*$ exists. Thus (17) and (18) imply that

$$\lim_{n\to\infty} R^\epsilon(f_n) = \lim_{n\to\infty} \int S_\epsilon(\mathbf{1}_{f_n \leq 0}) d\mathbb{P}_1 + \int S_\epsilon(\mathbf{1}_{f_n > 0}) d\mathbb{P}_0 = \lim_{n\to\infty} \int \mathbf{1}_{f_n \leq 0} d\mathbb{P}_1^* + \int \mathbf{1}_{f_n > 0} d\mathbb{P}_0^*$$

$$= \lim_{n\to\infty} \int C(\eta^*, f_n) d\mathbb{P}^* = \int C^*(\eta^*) d\mathbb{P}^* = \bar{R}(\mathbb{P}_0^*, \mathbb{P}_1^*).$$

Therefore, Strong duality (Theorem 1) then implies that

$$\lim_{n\to\infty} R^\epsilon(f_n) \leq \sup_{\substack{\mathbb{P}_0' \in \mathcal{B}_\epsilon^\infty(\mathbb{P}_0) \\ \mathbb{P}_1' \in \mathcal{B}_\epsilon^\infty(\mathbb{P}_1)}} \bar{R}(\mathbb{P}_0', \mathbb{P}_1') = \inf_{\substack{f \text{ Borel} \\ \mathbb{R}\text{-valued}}} R^\epsilon(f)$$

and therefore, $f_n$ is a minimizing sequence. $\square$

We end this section by comparing the different criteria for consistency presented in Proposition 1 with Propositions 4 and 5. Together, Propositions 4 and 5 will allow us to compare minimizing sequences of $R_\phi^\epsilon$ to those of $R^\epsilon$ by showing that any sequence satisfying (12)–(13) must also satisfy (17)–(18). This statement is the analog to 2) of Proposition 1. Indeed, because $C_\phi(\eta^*, f_n) \geq C_\phi^*(\eta^*)$, (12) is actually equivalent to to $C_\phi(\eta^*, f_n) \to C_\phi^*(\eta^*)$ in $L^1(\mathbb{P}^*)$. However, the extra criterion (18) implies an additional constraint on the structure of the minimizing sequence. This additional constraint is the reason 3) of Proposition 1 is false in the adversarial setting. In the restricted situation where $\bar{R}_\phi = \bar{R}$, Meunier et al. [2022] show that (12) implies (17) (Proposition 4.2). However, this observation does not suffice to conclude consistency.

## 4.2 Adversarial Consistency

We are now in a position to prove consistency. Before presenting the full proof, we pause to discuss the overall strategy. Consistency will follow from three considerations. First, every minimizing sequence of $R_\phi^\epsilon$ satisfies conditions (12) and (13). Second, conditions (12) and (13) imply the very similar conditions (17) and (18). Finally, any function sequence satisfying (17) and (18) must be a minimizing sequence to $R^\epsilon$. The first and last steps are the content of Propositions 4 and 5, so it remains to justify the middle step.

Verifying that (12) implies (17) is straightforward. The relation (12) actually states that $f_n$ minimizes the *standard* surrogate risk with respect to the distribution given by $\mathbb{P}_0^*, \mathbb{P}_1^*$. Therefore (12) implies (17) so long as $\phi$ is consistent.

The main difficulty is verifying (18), due to the discontinuity of $\mathbf{1}_{\alpha<0}, \mathbf{1}_{\alpha\geq0}$ at 0. Due to this discontinuity, one cannot directly argue that (13) implies (18): to simplify the discussion, assume that $\phi$ is strictly decreasing on a neighborhood of the origin, in which case $\mathbf{1}_{\alpha<0} = \mathbf{1}_{\phi(\alpha)>\phi(0)}$ and $\mathbf{1}_{\alpha\geq0} = \mathbf{1}_{\phi(-\alpha)\geq\phi(0)}$. Recall that according to (13), in the limit $n \to \infty$, $\mathbb{P}_0^*, \mathbb{P}_1^*$ are the strongest attack in $\mathcal{B}_\epsilon^\infty(\mathbb{P}_0) \times \mathcal{B}_\epsilon^\infty(\mathbb{P}_1)$, or informally, $S_\epsilon(\phi \circ f_n)(\mathbf{x})$ approaches $\phi(f_n(\mathbf{x}'))$ for an optimal perturbation $\mathbf{x}'$ w.h.p., with a similar condition for $\phi \circ -f_n$. However, due to the discontinuity of $\mathbf{1}_{\phi(-\alpha)\geq\phi(0)}$ at $\phi(0)$, if $f_n(\mathbf{x}') \to 0$ as $n \to \infty$, this relation does not imply that $\mathbf{1}_{S_\epsilon(\phi \circ -f_n)(\mathbf{x})\geq\phi(0)}$ approaches $\mathbf{1}_{\phi \circ -f_n(\mathbf{x}')\geq0}$.

Lemma 1 says that if $C_\phi^*(1/2) < \phi(0)$, minimizers of $C_\phi(\eta, \cdot)$ are uniformly bounded away from 0. This fact suggests that minimizing sequences will also be bounded away from the origin, which will allow us to avoid the discontinuity there. Concretely, we show:

**Lemma 3.** *Let* $C_\phi^*(1/2) < \phi(0)$. *Then there is a* $\delta > 0$ *and a* $c > 0$ *with* $\phi(c) < \phi(0)$ *for which* $\alpha \in [-c, c]$ *implies* $C_\phi(\eta, \alpha) \geq C_\phi^*(\eta) + \delta$, *uniformly in* $\eta$. *Furthermore, for this value of c, if* $\alpha > c$ *then* $\phi(\alpha) < \phi(c)$.

We prove this lemma in Appendix C. Because $C_\phi(\eta^*, f_n) \to C_\phi^*(\eta^*)$ in $L^1(\mathbb{P}^*)$, Lemma 3 implies that

$$\lim_{n\to\infty} \mathbb{P}^*(f_n \in [-c, c]) = 0. \tag{19}$$

This relation is the key fact that allows us to show that (13) implies (18). The condition $C_\phi^*(1/2) < \phi(0)$ is essential for this step of the argument.

Lastly, Lemma 2 implies that $\int S_\epsilon(\mathbf{1}_{f_n\geq0})d\mathbb{P}_1 \geq \int \mathbf{1}_{f_n\geq0}d\mathbb{P}_1^*$ and thus to validate (18), it suffices to verify the opposite inequality in the limit $n \to \infty$.

**Lemma 4.** *Let* $f_n$ *be a sequence of functions and let* $\mathbb{P}_0^* \in \mathcal{B}_\epsilon^\infty(\mathbb{P}_0)$, $\mathbb{P}_1^* \in \mathcal{B}_\epsilon^\infty(\mathbb{P}_1)$. *The equation*

$$\limsup_{n\to\infty} \int S_\epsilon(\mathbf{1}_{f_n\leq0})d\mathbb{P}_1 \leq \liminf_{n\to\infty} \int \mathbf{1}_{f_n\leq0}d\mathbb{P}_1^* \tag{20}$$

*implies the first relation of* (18) *and*

$$\limsup_{n\to\infty} \int S_\epsilon(\mathbf{1}_{f_n>0})d\mathbb{P}_0 \leq \liminf_{n\to\infty} \int \mathbf{1}_{f_n>0}d\mathbb{P}_0^* \tag{21}$$

*implies the second relation of* (18).

See Appendix F for a proof. These considerations suffice to prove the main result of this paper:

**Theorem 3.** *The loss $\phi$ is adversarially consistent if and only if $C_\phi^*(1/2) < \phi(0)$.*

*Proof.* The 'only if' portion of the statement is Proposition 2.

To show the 'if' statement, recall the standard analysis fact: $\lim_{n\to\infty} a_n = a$ iff for all subsequences $\{a_{n_j}\}$ of $\{a_n\}$, there is a further subsequence $a_{n_{j_k}}$ for which $\lim_{k\to\infty} a_{n_{j_k}} = a$. This result implies that to prove $R_\phi^\epsilon$ is consistent, it suffices to show that every minimizing sequence $f_n$ of $R_\phi^\epsilon$ has a subsequence $f_{n_j}$ that minimizes $R^\epsilon$.

Let $f_n$ be a minimizing sequence of $R_\phi^\epsilon$. For convenience, pick a subsequence $f_{n_j}$ for which the limits $\lim_{j\to\infty} \int S_\epsilon(\mathbf{1}_{f_{n_j}<0})d\mathbb{P}_0$, $\lim_{j\to\infty} \int S_\epsilon(\mathbf{1}_{f_{n_j}\geq 0})d\mathbb{P}_1$ both exist. For notational clarity, we drop the $_j$ subscript and denote this sequence as $f_n$.

By Proposition 4, the equations (12) and (13) hold. We will argue that $f_n$ is in fact a minimizing sequence of $R^\epsilon$ by verifying the conditions of Proposition 5.

First, the relation (12) states that the sequence $f_n$ minimizes the *standard $\phi$-risk* for the distribution given by $\mathbb{P}_0^*$ and $\mathbb{P}_1^*$. As the loss $\phi$ is consistent by Proposition 3, the sequence $f_n$ must minimize the standard classification risk for the distribution $\mathbb{P}_0^*, \mathbb{P}_1^*$. This statement implies (17). Next we will argue that (18) holds.

Let $c, \delta$ be as in Lemma 3. Because $C_\phi(\eta^*, f_n) \geq C_\phi^*(\eta^*)$, (12) implies that $C_\phi(\eta^*, f_n)$ converges to $C_\phi^*(\eta^*)$ in $L^1$. However, $L^1$ convergence implies convergence in measure (see for instance Proposition 2.29 of [Folland, 1999]), and therefore $\lim_{n\to\infty} \mathbb{P}^*\big(C_\phi(\eta^*, f_n) > C_\phi^*(\eta^*) + \delta\big) = 0$. Lemma 3 then implies that for $i = 0, 1$

$$\lim_{n\to\infty} \mathbb{P}_i^*(f_n \in [-c, c]) = 0. \tag{22}$$

Next, because $\phi$ is non-increasing, $f \leq 0$ implies $\phi(f) \geq \phi(0)$ and thus $\mathbf{1}_{f\leq 0} \leq \mathbf{1}_{\phi\circ f\geq\phi(0)}$. Furthermore, as the function $\alpha \mapsto \mathbf{1}_{\alpha\geq 0}$ is monotone and upper semi-continuous,

$$\int S_\epsilon(\mathbf{1}_{f_n\leq 0})d\mathbb{P}_1 \leq \int S_\epsilon(\mathbf{1}_{\phi\circ f_n\geq\phi(0)})d\mathbb{P}_1 \leq \int \mathbf{1}_{S_\epsilon(\phi\circ f_n)\geq\phi(0)}d\mathbb{P}_1. \tag{23}$$

Let $\gamma_i$ be a coupling between $\mathbb{P}_i$ and $\mathbb{P}_i^*$ for which $\text{ess}\sup_{(\mathbf{x},\mathbf{y})\sim\gamma_i} \|\mathbf{x} - \mathbf{y}\| \leq \epsilon$. Then the measure $\gamma_i$ is supported on $\Delta_\epsilon = \{(\mathbf{x}, \mathbf{y}) : \|\mathbf{x}-\mathbf{y}\| \leq \epsilon\}$. Furthermore, as $S_\epsilon(\phi\circ f_n)(\mathbf{x}) \geq \phi\circ f_n(\mathbf{x}')$ everywhere on $\Delta_\epsilon$, the relation $S_\epsilon(\phi\circ f_n)(\mathbf{x}) \geq \phi\circ f_n(\mathbf{x}')$ actually holds $\gamma_1$-a.e. Therefore, (13) actually implies that $S_\epsilon(\phi\circ f_n)(\mathbf{x}) - \phi\circ f_n(\mathbf{x}')$ converges in $\gamma_1$-measure to 0. In particular, since $\phi(c) < \phi(0)$, $\lim_{n\to\infty} \gamma_1\big(S_\epsilon(\phi\circ f_n)(\mathbf{x}) - \phi(f_n(\mathbf{x}')) \geq \phi(0) - \phi(c)\big) = 0$ and thus $\lim_{n\to\infty} \gamma_1(S_\epsilon(\phi\circ f_n)(\mathbf{x}) \geq \phi(0) \cap \phi\circ f_n(\mathbf{x}') < \phi(c)) = 0$. Therefore,

$$\liminf_{n\to\infty} \mathbb{P}_1(S_\epsilon(\phi\circ f_n)(\mathbf{x}) \geq \phi(0)) = \liminf_{n\to\infty} \gamma_1(S_\epsilon(\phi\circ f_n)(\mathbf{x}) \geq \phi(0) \cap \phi\circ f_n(\mathbf{x}') \geq \phi(c))$$
$$\leq \liminf_{n\to\infty} \gamma_1(\phi\circ f_n(\mathbf{x}') \geq \phi(c)) = \liminf_{n\to\infty} \mathbb{P}_1^*(\phi\circ f_n(\mathbf{x}') \geq \phi(c))$$

This calculation implies

$$\liminf_{n\to\infty} \int \mathbf{1}_{S_\epsilon(\phi\circ f_n)(\mathbf{x})\geq\phi(0)}d\mathbb{P}_1 \leq \liminf_{n\to\infty} \int \mathbf{1}_{\phi\circ f_n(\mathbf{x}')\geq\phi(c)}d\mathbb{P}_1^* \leq \liminf_{n\to\infty} \int \mathbf{1}_{f_n\leq c}d\mathbb{P}_1^* \tag{24}$$

The last inequality follows because Lemma 3 states that $\alpha > c$ implies $\phi(\alpha) < \phi(c)$ and therefore $\mathbf{1}_{\phi\circ f_n\geq\phi(c)} \leq \mathbf{1}_{f_n\leq c}$. Equation 22 then implies

$$\liminf_{n\to\infty} \int \mathbf{1}_{\phi\circ f_n\geq\phi(0)}d\mathbb{P}_1^* \leq \liminf_{n\to\infty} \int \mathbf{1}_{f_n\leq c}d\mathbb{P}_1^* = \liminf_{n\to\infty} \int \mathbf{1}_{f_n\leq -c}d\mathbb{P}_1^*. \tag{25}$$

Recall that the sequence $f_n$ was chosen so that the limit $\lim_{n\to\infty} \int S_\epsilon(\mathbf{1}_{f_n\leq 0})d\mathbb{P}_1$ exists. Combining this fact with (23), (24), and (25) results in

$$\limsup_{n\to\infty} \int S_\epsilon(\mathbf{1}_{f_n\leq 0})d\mathbb{P}_1 \leq \liminf_{n\to\infty} \int \mathbf{1}_{f_n\leq -c}d\mathbb{P}_1^* \leq \liminf_{n\to\infty} \int \mathbf{1}_{f_n\leq 0}d\mathbb{P}_1^* \tag{26}$$

The first relation of (18) then follows from (26) together with Lemma 4.

A similar argument implies the second relation of (18). Because $\mathbf{1}_{f>0} = \mathbf{1}_{-f<0} \leq \mathbf{1}_{-f\leq 0}$, the same chain of inequalities as (23), (24), and (25) implies that

$$\limsup_{n\to\infty} \int S_\epsilon(\mathbf{1}_{f_n>0})d\mathbb{P}_0 \leq \limsup_{n\to\infty} \int S_\epsilon(\mathbf{1}_{-f_n\leq 0})d\mathbb{P}_0 \leq \liminf_{n\to\infty} \int \mathbf{1}_{-f_n\leq -c}d\mathbb{P}_0^* = \liminf_{n\to\infty} \int \mathbf{1}_{f_n\geq c}d\mathbb{P}_0^*$$

As $c > 0$, it follows that $\limsup_{n\to\infty} \int S_\epsilon(\mathbf{1}_{f_n>0})d\mathbb{P}_0 \leq \liminf_{n\to\infty} \int \mathbf{1}_{f_n>0}d\mathbb{P}_0^*$. Once again, the second expression of (18) follows from this relation and Lemma 4. □

## 5 Quantitative Bounds for the $\rho$-Margin Loss

As discussed in the introduction, statistical consistency is not the only property one would want from a surrogate. Hopefully, minimizing a surrogate will also efficiently minimize the classification loss. Bartlett et al. [2006], Steinwart [2007], Reid and Williamson [2009] prove bounds of the form $R(f) - R_* \leq G_\phi(R_\phi^*(f) - R_{\phi,*})$ for a function $G_\phi$ and $R_* = \inf_f R(f)$, $R_{\phi,*} = \inf_f R_\phi(f)$. The function $G_\phi$ is an upper bound on the rate of convergence of the classification risk in terms of the rate of convergence of the surrogate risk. One would hope that $G_\phi$ is not logarithmic, as such a bound could imply that reducing $R(f) - R_*$ by a quantity $\Delta$ could require an exponential change of $e^\Delta$ in $R_\phi(f) - R_{\phi,*}$. Bartlett et al. [2006] compute such $G_\phi$ for several popular losses in the standard classification setting. For example, they show the bounds $G_\phi(\theta) = \theta$ for the hinge loss $\phi(\alpha) = (1 - \alpha)_+$ and $G_\phi(\theta) = \sqrt{\theta}$ for the squared hinge loss $\phi(\alpha) = (1 - \alpha)_+^2$. On can prove an analogous bound for the $\rho$-margin loss in the adversarial setting:

**Theorem 4.** *Let* $\phi_\rho = \min(1, \max(1 - \alpha/\rho, 0))$ *be the $\rho$-margin loss,* $R_*^\epsilon = \inf_f R^\epsilon(f)$*, and* $R_{\phi_\rho,*}^\epsilon(f) = \inf_f R_{\phi_\rho}^\epsilon(f)$*. Then*

$$R^\epsilon(f) - R_*^\epsilon \leq R_{\phi_\rho}^\epsilon(f) - R_{\phi_\rho,*}^\epsilon.$$

Notice that this theorem immediately implies that the $\rho$-margin loss is in fact adversarially consistent. The proof below is completely independent of the argument in Section 4.

*Proof.* Notice that for the $\rho$-margin loss, $C_{\phi_\rho}^* = C^*$ and therefore, the optimal $\phi_\rho$-risk $R_{\phi_\rho,*}^\epsilon$ equals the optimal adversarial classification risk $R_*^\epsilon$. However, since $\phi_\rho(\alpha) \geq \mathbf{1}_{\alpha\leq 0}$ and $\phi_\rho(-\alpha) \geq \mathbf{1}_{\alpha>0}$ for any $\alpha$, one can conclude that $R^\epsilon(f) \leq R_{\phi_\rho}^\epsilon(f)$. Therefore,

$$R^\epsilon(f) - R_*^\epsilon = R^\epsilon(f) - R_{\phi_\rho,*}^\epsilon \leq R_{\phi_\rho}^\epsilon(f) - R_{\phi_\rho,*}^\epsilon$$

□

This bound implies that reducing the excess adversarial $\rho$-margin loss by $\Delta$ also reduces an upper bound on the excess adversarial classification loss by $\Delta$. Thus, one would expect that minimizing the adversarial $\rho$-margin risk would be an effective procedure for minimizing the adversarial classification risk.

Extending Theorem 4 to other losses remains an open problem. In the non-adversarial scenario, many prior works develop techniques for computing such bounds. These include the $\Psi$-transform of [Bartlett et al., 2006], calibration analysis in [Steinwart, 2007], and special techniques for proper losses in [Reid and Williamson, 2009].

Contemporary work [Mao et al., 2023] derives an $\mathcal{H}$-consistency surrogate risk bound for a variant of the adversarial $\rho$-margin loss.

## 6 Conclusion

In conclusion, we proved that the adversarial training procedure is consistent for perturbations in an $\epsilon$-ball if an only if $C_\phi^*(1/2) < \phi(0)$. The technique that proved consistency extends to perturbation sets which satisfy existence and minimax theorems analogous to Theorems 1 and 2. Furthermore, we showed a quantitative excess risk bound for the adversarial $\rho$-margin loss. Finding such bounds for other losses remains an open problem. We hope that insights to consistency and the structure of adversarial learning will lead to the design of better adversarial learning algorithms.

## Acknowledgments and Disclosure of Funding

Natalie Frank was supported in part by the Research Training Group in Modeling and Simulation funded by the National Science Foundation via grant RTG/DMS – 1646339. Jonathan Niles-Weed was supported in part by a Sloan Research Fellowship.

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
