# A   An Alternative Characterization of Consistency– Proof of Proposition 1

First, prior work computes the minimum standard $\phi$-risk.

**Lemma 5.** *Let $\phi$ be any monotonic loss function. Then*

$$\inf_{f \text{ measurable}} R_\phi(f) = \int C_\phi^*(\eta) d\mathbb{P}$$

This result appears on page 4 of [Bartlett et al., 2006]. Notice that Lemma 5 is Theorem 2 with $\epsilon = 0$. Next, one can use the following lemma to compare minimizing sequences of $C_\phi(\eta, \cdot)$ and $C(\eta, \cdot)$.

**Lemma 6.** *Assume that Assumption 1 holds, $\phi$ is consistent, and $0 \in \operatorname{argmin} C_\phi(\eta, \cdot)$. Then $\eta = 1/2$.*

*Proof.* Consider a distribution for which $\eta(\mathbf{x}) \equiv \eta$ is constant. Then $R_\phi(f) = C_\phi(\eta, f)$ and $R(f) = C(\eta, f)$. The consistency of $\phi$ implies that if $0$ minimizes $C_\phi(\eta, \cdot)$, then it also must minimize $C(\eta, \cdot)$ and therefore $\eta \leq 1/2$.

However, notice that $C_\phi(\eta, \alpha) = C_\phi(1 - \eta, -\alpha)$. Thus if $0$ minimizes $C_\phi(\eta, \cdot)$ it must also minimize $C_\phi(1 - \eta, \cdot)$. The consistency of $\phi$ then implies that $1 - \eta \leq 1/2$ as well and consequently, $\eta = 1/2$. $\square$

We use this result to prove Proposition 1 together with a standard argument from analysis:

**Lemma 7.** *Let $\{a_n\}$ be a sequence in $\mathbb{R} \cup \{\infty\}$. Then the following are equivalent:*

   *1) $\lim_{n \to \infty} a_n = a$*

   *2) Every subsequence $\{a_{n_j}\}$ of $\{a_n\}$ has a subsequence $\{a_{j_k}\}$ for which $\lim_{k \to \infty} a_{j_k} = a$*

As a result:

**Corollary 1.** *If every minimizing sequence $f_n$ of $R_\phi$ has a subsequence $f_{n_j}$ that minimizes $R$, then $\phi$ is consistent.*

Furthermore, this corollary can be applied to a distribution with constant $\eta(\mathbf{x})$ to conclude:

**Corollary 2.** *If every minimizing sequence $\alpha_n$ for $C_\phi(\eta, \cdot)$ has a subsequence $\alpha_{n_j}$ that minimizes $C(\eta, \cdot)$ then one can conclude that every minimizing sequence is of $C_\phi(\eta, \cdot)$ is also a minimizing sequence of $C(\eta, \cdot)$.*

We now prove a result slightly stronger than Proposition 1.

**Theorem 5.** *The following are equivalent:*

   *1) For all distributions, $f_n$ is a minimizing sequence of $R_\phi$ implies that $f_n$ is a minimizing sequence of $R$.*

   *2) For all $\eta \in [0, 1]$, $\alpha_n$ is a minimizing sequence of $C_\phi(\eta, \cdot)$ implies that $\alpha_n$ is a minimizing sequence of $C(\eta, \cdot)$.*

   *3) Every minimizer of $C_\phi(\eta, \cdot)$ is also a minimizer of $C(\eta, \cdot)$.*

   *4) Every minimizer of $R_\phi$ is a minimizer of $R$*

The proof is essentially the "pointwise" argument discussed in Section 3.

*Proof.* We show that 1) $\Leftrightarrow$ 2), 2) $\Leftrightarrow$ 3), and 3) $\Leftrightarrow$ 4).

**Showing 1) is equivalent to 2):**

To show that 1) implies 2), consider a distribution for which $\eta(\mathbf{x}) \equiv \eta$ is constant.

For the other direction, let $f_n$ be any minimizing sequence of $R_\phi$. Then $C_\phi(\eta, f_n) \geq C_\phi^*(\eta)$ and Lemma 5 implies that the sequence $C_\phi(\eta, f_n)$ actually converges to $C_\phi^*(\eta)$ in $L^1(\mathbb{P})$. Thus one

can pick a subsequence $f_{n_j}$ for which $C_\phi(\eta, f_{n_j})$ converges to $C_\phi^*(\eta)$ $\mathbb{P}$-a.e. (See for instance Corollary 2.32 of [Folland, 1999]). Then 2) implies that the function sequence $f_{n_j}$ minimizes $C(\eta, \cdot)$ and therefore it also minimizes $R$ by Corollary 1.

**Showing 2) is equivalent to 3):**

To show that 2) implies 3), notice that if $\alpha$ is a minimizer of $C_\phi(\eta, \cdot)$, 2) immediately implies that the sequence $\alpha_n \equiv \alpha$ also minimizes $C(\eta, \cdot)$.

For the other direction, assume that every minimizer of $C_\phi(\eta, \cdot)$ is also a minimizer of $C(\eta, \cdot)$. Let $\alpha_n$ be a minimizing sequence of $C_\phi(\eta, \cdot)$. Over the extended real numbers $\overline{\mathbb{R}}$, $\alpha_n$ has a subsequence $\alpha_{n_j}$ that converges to a limit point $a$, which must be a minimizer of $C_\phi(\eta, \cdot)$. Now if $a \neq 0$, both $\mathbf{1}_{\alpha \leq 0}, \mathbf{1}_{\alpha > 0}$ are continuous at $a$ so that one can conclude that $\alpha_{n_j}$ also minimizes $C(\eta, \cdot)$. If in fact $a = 0$, Lemma 6 implies that $\eta = 1/2$ and thus *any* $\alpha$ minimizes $C(1/2, \cdot)$. Thus Corollary 2 implies that $\alpha_n$ minimizes $C(\eta, \cdot)$.

**Showing 3) is equivalent to 4)**

To show that 4) implies 3), consider a distribution for which $\eta(\mathbf{x}) \equiv \eta$ is constant.

For the other direction, let $f^*$ be a minimizer of $R_\phi$. Then $C_\phi(\eta(\mathbf{x}), f^*(\mathbf{x})) \geq C_\phi^*(\eta(\mathbf{x}))$ but $R_\phi(f^*) = \int C_\phi^*(\eta) d\mathbb{P}$ by Lemma 5. Therefore $C_\phi(\eta(\mathbf{x}), f^*(\mathbf{x})) = C_\phi^*(\eta(\mathbf{x}))$ $\mathbb{P}$-a.e. Item 3) then implies the result. $\qquad\square$

# B    Minimizing $R_\phi^\epsilon$ over $\overline{\mathbb{R}}$-valued functions

In this appendix, we will show

**Lemma 8.** *Let $R_\phi^\epsilon$ be defined as in (7). Then*

$$\inf_{\substack{f\ Borel,\\ f\ \mathbb{R}\text{-valued}}} R_\phi^\epsilon(f) = \inf_{\substack{f\ Borel,\\ f\ \overline{\mathbb{R}}\text{-valued}}} R_\phi^\epsilon(f)$$

Integrals of functions assuming values in $\mathbb{R} \cup \{\infty\}$ can still be defined using standard measure theory, see for instance [Folland, 1999].

Recall that [Frank and Niles-Weed, 2023] originally proved their minimax result for $\overline{\mathbb{R}}$-valued functions and thus this lemma is essential for the statement of Theorem 2.

*Proof of Lemma 8.* Let $f$ be an $\overline{\mathbb{R}}$-valued function for with $R_\phi^\epsilon(f) < \infty$. We will show that the truncation $f_N = \min(\max(f, -N), N)$ satisfies $\lim_{N \to \infty} R_\phi^\epsilon(f_N) = R_\phi^\epsilon(f)$. Lemma 8 then follows from this statement.

Define a function $\sigma_{[a,b]} \colon \overline{\mathbb{R}} \to [a, b]$ by

$$\sigma_{[a,b]}(\alpha) = \begin{cases} b & \alpha > b \\ \alpha & \alpha \in [a, b] \\ a & \alpha < a \end{cases}$$

Notice that $\sigma_{[a,b]}(-\alpha) = -\sigma_{[-b,-a]}(\alpha)$. Thus if $a = -b$, then $\sigma_{[a,b]}$ is anti-symmetric. Furthermore, because $\phi$ is continuous and non-increasing, for any function $g$,

$$\phi(\sigma_{[a,b]}(g)) = \sigma_{[\phi(b), \phi(a)]}(\phi(g))$$

and as $\sigma_{[a,b]}(\alpha)$ is continuous and non-decreasing,

$$S_\epsilon(\sigma_{[a,b]}(g)) = \sigma_{[a,b]}(S_\epsilon(g))$$

Now let $f_N = \sigma_{[-N,N]}(f)$. Then $S_\epsilon(\phi \circ f_N), S_\epsilon(\phi \circ -f_N)$ satisfy

$$S_\epsilon(\phi(f_N)) = \sigma_{[\phi(N), \phi(-N)]}(S_\epsilon(\phi \circ f)), \quad S_\epsilon(\phi(-f_N)) = \sigma_{[\phi(N), \phi(-N)]}(S_\epsilon(\phi \circ -f))$$

Therefore, $S_\epsilon(\phi \circ f_N)$, $S_\epsilon(\phi \circ -f_N)$ converge pointwise to $S_\epsilon(\phi \circ f)$, $S_\epsilon(\phi \circ -f)$. Furthermore, for $N \geq 1$, $\phi(f_N) \leq \phi(f) + \phi(1)$ which is integrable with respect to $\mathbb{P}_1$. Similarly, $\phi(-f_N) \leq \phi(-f) + \phi(1)$ which is integrable with respect to $\mathbb{P}_0$. Therefore, the dominated convergence theorem implies that

$$\lim_{N \to \infty} R_\phi^\epsilon(f_N) = R_\phi^\epsilon(f)$$

$\square$

## C  Further Properties of Adversarially Consistent Losses– Proofs of Lemma 1, Lemma 3, and Proposition 3

Recall the condition $C_\phi^*(1/2) < \phi(0)$ implies that minimizers of $C_\phi(1/2, \alpha)$ are bounded away from zero. Lemma 9 states that this property actually holds for *all* $\eta$. To prove this fact, we decompose $C_\phi(\eta, \alpha)$ into $C_\phi(1/2, \alpha)$ and a monotonic function:

$$C_\phi(\eta, \alpha) = \eta\phi(\alpha) + (1-\eta)\phi(-\alpha) = (\eta - 1/2)(\phi(\alpha) - \phi(-\alpha)) + \frac{1}{2}(\phi(\alpha) + \phi(-\alpha)). \quad (27)$$

**Lemma 9.** *Assume that $C_\phi^*(1/2) < \phi(0)$. Then there exists an $a > 0$ for which $|\alpha| < a$ implies $C_\phi(\eta, \alpha) \neq C_\phi^*(\eta)$ for all $\eta$. This $a$ satisfies $\phi(a) < \phi(0)$.*

*Proof.* Let $S$ be the set of non-negative minimizers of $C_\phi(1/2, \cdot)$ and define $a = \inf S$. Because $\phi$ is continuous, $a$ is also a minimizer of $C_\phi(1/2, \cdot)$ and thus $C_\phi(1/2, a) = C_\phi^*(1/2) < \phi(0) = C_\phi(1/2, 0)$. Therefore, $\phi(a) < \phi(0)$ follows from the fact that $\phi(-a) \geq \phi(0)$.

We will now show that $C_\phi(\eta, \cdot)$ does not achieve its optimum on $(-a, a)$ for any $\eta$. First, this statement holds for $\eta = 1/2$ due to the definition of $a$. Next, we will assume that $\eta > 1/2$, the case $\eta < 1/2$ is analogous. To start, we can decompose the quantity $C_\phi(\eta, \alpha)$ as in (27). Subsequently, because $a$ is the smallest positive minimizer of $C_\phi(1/2, \cdot)$, $1/2(\phi(\alpha) + \phi(-\alpha))$ assumes its infimum over $[-a, a]$ only at $-a$ and $a$. Next, notice that $\phi(\alpha) - \phi(-\alpha)$ is non-increasing on $[-a, a]$. Furthermore, because $\phi(a) < \phi(0)$, one can conclude that $\phi(-a) - \phi(a) > 0 > \phi(a) - \phi(-a)$, and thus the function $\alpha \mapsto \phi(\alpha) - \phi(-\alpha)$ is non-constant on $[-a, a]$. Therefore, (27) achieves its optimum over $[-a, a]$ only at $\alpha = a$. Thus, any $\alpha \in (-a, a)$ cannot be a minimizer of $C_\phi(\eta, \cdot)$ because $C_\phi(\eta, \alpha) > C_\phi(\eta, a) \geq C_\phi^*(\eta)$.

$\square$

*Proof of Lemma 1.* Lemma 9 (above) immediately implies the forward direction.

For the backwards direction, note that if there is an $a$ for which $|\alpha^*| \geq a$ for any minimizer $C_\phi(\eta, \cdot)$ for all $\eta$, then $0$ does not minimize $C_\phi(1/2, \cdot)$. Therefore $C_\phi^*(1/2) < C_\phi(1/2, 0) = \phi(0)$.

$\square$

*Proof of Proposition 3.* We will argue that for each $\eta$, every minimizer of $C_\phi(\eta, \cdot)$ over $\overline{\mathbb{R}}$ is also a minimizer of $C(\eta, \cdot)$. Proposition 1 will then imply that $\phi$ is consistent. To start, notice that *every* $\alpha$ is a minimizer of $C(1/2, \cdot)$. Next, we will show that for $\eta > 1/2$, every minimizer of $C_\phi(\eta, \cdot)$ is also a minimizer of $C(\eta, \cdot)$. The argument for $\eta < 1/2$ is analogous.

Consider the decomposition of $C_\phi(\eta, \alpha)$ in (27). Let $a$ be as in Lemma 9 and notice that if $\alpha > a$ then $\phi(\alpha) < \phi(-\alpha)$. Hence as $\eta > 1/2$, then $C_\phi(\eta, \alpha) < C_\phi(\eta, -\alpha)$. Furthermore, Lemma 9 implies that there is no minimizer to $C_\phi(\eta, \cdot)$ in $(-a, a)$ and thus every minimizer to $C_\phi(\eta, \cdot)$ must be strictly positive. Therefore, every minimizer of $C_\phi(\eta, \cdot)$ also minimizes $C(\eta, \cdot)$.

$\square$

Next, Lemma 3 is a quantitative version of Lemma 9.

*Proof of Lemma 3.* Let $a$ be as in Lemma 9 and define $\phi^-$ by

$$\phi^-(y) = \sup\{\alpha : \phi(\alpha) \geq y\}.$$

The function $\phi^-$ is the right inverse of $\phi$— this function satisfies $\phi(\phi^-(y)) = y$ while $\phi^-(\phi(\alpha)) \geq \alpha$.

Set $k = 1/2(\phi(0) + \phi(a))$, $c = \phi^-(k) = \sup\{\alpha: \phi(\alpha) \geq k\}$. From the definition of $c$, one can conclude that $\alpha > c$ implies that $\phi(\alpha) < \phi(c)$.

Because $\phi(a) < k = \phi(c) < \phi(0)$ and $\phi$ is non-increasing, $0 < c < a$. Thus $[-c, c] \subset (-a, a)$ and Lemma 9 implies that for all $\alpha \in [-c, c]$ and $\eta \in [0, 1]$, $C_\phi(\eta, \alpha) - C_\phi^*(\eta) > 0$. As this expression is jointly continuous in the variables $\eta$, $\alpha$ and $[-c, c] \times [0, 1]$ is compact, one can define

$$\delta = \inf_{\substack{\alpha \in [-c,c] \\ \eta \in [0,1]}} C_\phi(\eta, \alpha) - C_\phi^*(\eta)$$

and then it holds that $\delta > 0$ and $C_\phi(\eta, \alpha) \geq C_\phi^*(\eta) + \delta$ for all $\alpha \in [-c, c]$.

$\square$

# D    Optimal Transport Facts— Proof of Lemma 2

*Proof of Lemma 2.* Let $\mathbb{Q}'$ be any measure with $W_\infty(\mathbb{Q}', \mathbb{Q}) \leq \epsilon$. Let $\gamma$ be a coupling with marginals $\mathbb{Q}$ and $\mathbb{Q}'$ for which $\operatorname{ess\,sup}_{(\mathbf{x},\mathbf{y}) \sim \gamma} \|\mathbf{x} - \mathbf{y}\| \leq \epsilon$. This measure $\gamma$ is supported on $\Delta_\epsilon = \{(\mathbf{x}, \mathbf{y}): \|\mathbf{x} - \mathbf{y}\| \leq \epsilon\}$. Then

$$\int g d\mathbb{Q}' = \int g(\mathbf{x}') d\gamma(\mathbf{x}, \mathbf{x}') = \int g(\mathbf{x}') \mathbf{1}_{\|\mathbf{x}' - \mathbf{x}\| \leq \epsilon} d\gamma(\mathbf{x}, \mathbf{x}')$$

$$\leq \int S_\epsilon(g)(\mathbf{x}) \mathbf{1}_{\|\mathbf{x}' - \mathbf{x}\| \leq \epsilon} d\gamma(\mathbf{x}, \mathbf{x}') = \int S_\epsilon(g)(\mathbf{x}) d\gamma(\mathbf{x}, \mathbf{x}') = \int S_\epsilon(g) d\mathbb{Q}$$

$\square$

# E    Proof of Theorem 1

As observed in Section 5, the $\rho$-margin loss satisfies $R_{\phi_\rho}^\epsilon(f) \geq R^\epsilon(f)$ while $C_{\phi_\rho}^*(\eta) = C^*(\eta)$. Theorem 2 then implies that

$$\sup_{\substack{\mathbb{P}_0' \in \mathcal{B}_\epsilon^\infty(\mathbb{P}_0) \\ \mathbb{P}_1' \in \mathcal{B}_\epsilon^\infty(\mathbb{P}_1)}} \bar{R}(\mathbb{P}_0', \mathbb{P}_1') = \sup_{\substack{\mathbb{P}_0' \in \mathcal{B}_\epsilon^\infty(\mathbb{P}_0) \\ \mathbb{P}_1' \in \mathcal{B}_\epsilon^\infty(\mathbb{P}_1)}} \bar{R}_{\phi_\rho}(\mathbb{P}_0', \mathbb{P}_1') = \inf_f R_{\phi_\rho}^\epsilon(f) \geq \inf_f R^\epsilon(f)$$

The opposite inequality follows from swapping an $\inf$ and a $\sup$— a form of weak duality. We prove this weak duality for $\overline{\mathbb{R}} = \mathbb{R} \cup \{-\infty, +\infty\}$-valued functions in order to later apply a result from [Frank and Niles-Weed, 2023] which is also stated for $\overline{\mathbb{R}}$-valued functions.

**Lemma 10** (Weak Duality). *Let $R^\epsilon$ be the adversarial classification loss. Then*

$$\inf_{\substack{f \text{ Borel,} \\ f \,\overline{\mathbb{R}}\text{-valued}}} R^\epsilon(f) \geq \sup_{\substack{\mathbb{P}_0' \in \mathcal{B}_\epsilon^\infty(\mathbb{P}_0) \\ \mathbb{P}_1' \in \mathcal{B}_\epsilon^\infty(\mathbb{P}_1)}} \bar{R}(\mathbb{P}_0', \mathbb{P}_1') \tag{28}$$

*Proof.* Notice that Lemma 2 implies that for any function $g$,

$$\int S_\epsilon(g) d\mathbb{Q} \geq \sup_{\mathbb{Q}' \in \mathcal{B}_\epsilon^\infty(\mathbb{Q})} \int g d\mathbb{Q}'.$$

Applying this inequality to the functions $\mathbf{1}_{f \leq 0}, \mathbf{1}_{f > 0}$ in the expression for $R^\epsilon(f)$ results in

$$\int S_\epsilon(\mathbf{1}_{f \leq 0}) d\mathbb{P}_1 + \int S_\epsilon(\mathbf{1}_{f > 0}) d\mathbb{P}_0 \geq \sup_{\substack{\mathbb{P}_0' \in \mathcal{B}_\epsilon^\infty(\mathbb{P}_0) \\ \mathbb{P}_1' \in \mathcal{B}_\epsilon^\infty(\mathbb{P}_1)}} \int \mathbf{1}_{f \geq 0} d\mathbb{P}_1' + \int \mathbf{1}_{f < 0} d\mathbb{P}_0'$$

Thus by swapping the $\inf$ and the $\sup$ and defining $\mathbb{P}' = \mathbb{P}'_0 + \mathbb{P}'_1$, $\eta' = d\mathbb{P}'_1/d\mathbb{P}'$,

$$\inf_{\substack{f \text{ Borel} \\ f \ \mathbb{R}\text{-valued}}} \int S_\epsilon(\mathbf{1}_{f \leq 0})d\mathbb{P}_1 + \int S_\epsilon(\mathbf{1}_{f > 0})d\mathbb{P}_0 \geq \inf_{\substack{f \text{ Borel} \\ f \ \mathbb{R}\text{-valued}}} \sup_{\substack{\mathbb{P}'_0 \in \mathcal{B}^\infty_\epsilon(\mathbb{P}_0) \\ \mathbb{P}'_1 \in \mathcal{B}^\infty_\epsilon(\mathbb{P}_1)}} \int \mathbf{1}_{f \leq 0}d\mathbb{P}'_1 + \int \mathbf{1}_{f > 0}d\mathbb{P}'_0$$

$$\geq \sup_{\substack{\mathbb{P}'_0 \in \mathcal{B}^\infty_\epsilon(\mathbb{P}_0) \\ \mathbb{P}'_1 \in \mathcal{B}^\infty_\epsilon(\mathbb{P}_1)}} \inf_{\substack{f \text{ Borel} \\ f \ \mathbb{R}\text{-valued}}} \int \mathbf{1}_{f \leq 0}d\mathbb{P}'_1 + \int \mathbf{1}_{f > 0}d\mathbb{P}'_0$$

$$= \sup_{\substack{\mathbb{P}'_0 \in \mathcal{B}^\infty_\epsilon(\mathbb{P}_0) \\ \mathbb{P}'_1 \in \mathcal{B}^\infty_\epsilon(\mathbb{P}_1)}} \inf_{\substack{f \text{ Borel} \\ f \ \overline{\mathbb{R}}\text{-valued}}} \int C(\eta', f)d\mathbb{P}' \geq \sup_{\substack{\mathbb{P}'_0 \in \mathcal{B}^\infty_\epsilon(\mathbb{P}_0) \\ \mathbb{P}'_1 \in \mathcal{B}^\infty_\epsilon(\mathbb{P}_1)}} \int C^*(\eta')d\mathbb{P}' = \sup_{\substack{\mathbb{P}'_0 \in \mathcal{B}^\infty_\epsilon(\mathbb{P}_0) \\ \mathbb{P}'_1 \in \mathcal{B}^\infty_\epsilon(\mathbb{P}_1)}} \bar{R}(\mathbb{P}'_0, \mathbb{P}'_1)$$

$\square$

Strong duality and existence of maximizers/minimizers then follows from weak duality.

*Proof of Theorem 1.* Let $\phi_\rho(\alpha)$ be the $\phi$-margin loss $\phi_\rho = \min(1, \max(1 - \alpha/\rho, 0))$. Then as discussed in Section 5, one can bound the adversarial classification risk $R^\epsilon(f)$ by $R^\epsilon(f) \leq R^\epsilon_{\phi_\rho}(f)$ but $C^*_{\phi_\rho}(\eta) = C^*(\eta)$ and thus $\bar{R}_{\phi_\rho} = \bar{R}$.

The minimax theorem for surrogate losses in [Frank and Niles-Weed, 2023] (Theorem 6) states that there is an $\overline{\mathbb{R}}$-valued function $f^*$, and measures $\mathbb{P}^*_0, \mathbb{P}^*_1$ for which $R^\epsilon_{\phi_\rho}(f^*) = \bar{R}_{\phi_\rho}(\mathbb{P}^*_0, \mathbb{P}^*_1)$. Thus weak duality (Lemma 10) implies

$$\bar{R}_{\phi_\rho}(\mathbb{P}^*_0, \mathbb{P}^*_1) = \bar{R}(\mathbb{P}^*_0, \mathbb{P}^*_1) \leq R^\epsilon(f^*) \leq R^\epsilon_{\phi_\rho}(f^*).$$

However, the fact that $R^\epsilon_{\phi_\rho}(f^*) = \bar{R}_{\phi_\rho}(\mathbb{P}^*_0, \mathbb{P}^*_1)$ implies that the inequalities above must actually be equalities. This relation proves strong duality for the adversarial classification risk (Equation 9) and that $f^*$ minimizes $R^\epsilon$ and $(\mathbb{P}^*_0, \mathbb{P}^*_1)$ maximizes $\bar{R}$ over $\mathcal{B}^\infty_\epsilon(\mathbb{P}_0) \times \mathcal{B}^\infty_\epsilon(\mathbb{P}_1)$.

Next, let $\hat{f} = \min(1, \max(\hat{f}, -1))$. Then $\hat{f}$ is $\mathbb{R}$-valued and $R^\epsilon(\hat{f}) = R^\epsilon(f^*)$. Thus $\hat{f}$ is an $\mathbb{R}$-valued minimizer of $R^\epsilon$. $\square$

## F  Proof of Lemma 4

*Proof of Lemma 4.* Lemma 2 implies that for each $n$,

$$\int S_\epsilon(\mathbf{1}_{f_n \leq 0})d\mathbb{P}_1 \geq \int \mathbf{1}_{f_n \leq 0}d\mathbb{P}^*_1.$$

Therefore, writing $\ell_n = \int S_\epsilon(\mathbf{1}_{f_n \leq 0})d\mathbb{P}_1$ and $r_n = \int \mathbf{1}_{f_n \leq 0}d\mathbb{P}^*_1$, we have

$$\liminf_{n \to \infty} r_n \leq \liminf_{n \to \infty} \ell_n \leq \limsup_{n \to \infty} \ell_n. \tag{29}$$

Therefore, (20) implies both that that the limit $\lim_{n \to \infty} \int S_\epsilon(\mathbf{1}_{f_n \leq 0})d\mathbb{P}_1$ exists and that

$$\lim_{n \to \infty} \int S_\epsilon(\mathbf{1}_{f_n \leq 0})d\mathbb{P}_1 = \liminf_{n \to \infty} \int \mathbf{1}_{f_n \leq 0}d\mathbb{P}^*_1 \tag{30}$$

Similarly, because $\limsup_{n \to \infty} \ell_n \geq \limsup_{n \to \infty} r_n \geq \liminf_{n \to \infty} r_n$, the relation (20) implies that the limit $\lim_{n \to \infty} \int \mathbf{1}_{f_n \leq 0}d\mathbb{P}^*_1$ exists. The first relation of (18) then follows from (30) and the existence of the limit of $\int \mathbf{1}_{f_n \leq 0}d\mathbb{P}^*_1$.

An analogous argument shows that (21) implies the second relation of (18). $\square$