# OpenReview forum: "The Adversarial Consistency of Surrogate Risks for Binary Classification"
_NeurIPS.cc/2023/Conference — NeurIPS 2023 poster_

### Official Review · Reviewer_QDX6 · 2023-06-16

**Soundness:** 3 good
**Presentation:** 3 good
**Contribution:** 3 good
**Rating:** 8
**Confidence:** 3

**Summary:**

This paper proves the necessary and sufficient condition for a loss function to be adversarially consistent.
In the previous literature, either adversarial consistency for restricted hypothesis spaces or negative results for adversarial consistency has been known.
This paper follows this research line to provide a general condition to characterize loss functions.
The condition only requires $C\_\\phi^\*(1/2) < \\phi(0)$, which is quite simple to check.
This even holds for nonconvex loss functions.
The proof technique relies on the strong duality and complementary slackness results between adversarial surrogate loss minimization and the optimal coupling between benign and adversarial distributions.

**Strengths:**

- Very general condition to characterize adversarially consistent losses: Unlike previous results of adversarial $\\mathcal{H}$-consistency (Awasthi et al. (2021)) and negative results for adversarial consistency (Meunier et al. (2022)), this work contributes to show the necessary and sufficient condition for a loss function to be adversarially consistent. This is a new insight into the community and may help design loss functions in adversarial training.
- The condition applies even to nonconvex losses: Traditionally, the theory of calibrated losses mainly concerns convex losses, such as Bartlett et al. (2006) because their proof technique essentially relies on the first-order optimality condition when characterizing loss minimizers. This is a transparent proof technique yet excludes nonconvex losses. In contrast, the proof technique of this paper first translates the optimality of the adversarial loss into the optimal coupling (Propositions 4 and 5) and then deals with the standard consistency analysis for the adversarial distribution $\\mathbb{P}^\*$ (by leveraging Lemma 1).

**Weaknesses:**

Overall, I do not see any concerns about this paper.
There are a few minor comments and questions, which are mentioned in the following "Questions."

**Questions:**

- (Comment) In the introduction, you may consider emphasizing the main result shown in this paper is related to consistency for all measurable functions, not $\\mathcal{H}$-consistency, to clarify how the result differs from the previous works.
- (Question) Regarding Proposition 2: The counterexample $\\mathbb{P}\_0 = \\mathbb{P}\_1$ seems very malicious and rarely happens in practice. Are there any other counterexamples for which the corresponding loss function is not adversarially consistent?
- (Comment) In the definition of $W\_\\infty$, it is better to explain what $(x,y) \\sim \\gamma$ does mean.
- (Typo) In l.233 "fo" -> "of"
- (Typo) In l.260 $R$ -> $R^\\epsilon$
- (Comment) In l.275, it seems better to discuss the existence of the coupling $\\gamma\_i$.
- (Comment) In l.307, I don't think Theorem 4 immediately implies adversarial consistency because it is unclear whether the inequality is tight for any distributions.

**Limitations:**

The authors mention the limitations in conclusion: The extension of the convergence rate for general loss functions is left open.

This is theoretical work, and potential negative societal concerns are not applicable.

---

> ### Author Rebuttal · Authors · 2023-08-08
>
> - Thank you for catching those typos. We have corrected them in our paper.
>
>
> - Regarding proposition 2: Yes you are correct that the counter example $\mathbb P_0=\mathbb P_1$ is particularly malicious. We are currently writing a follow-up paper that identifies all the counterexamples to consistency for $\phi$ satisfying $C_\phi^*(1/2)=\phi(0)$.
>
>
> - About Theorem 4 and consistency: Let $f_n$ be a minimizing sequence of $R_{\phi_\rho}^\epsilon$: then $\lim_{n\to \infty} R_{\phi_\rho}^\epsilon(f_n)=R^\epsilon_{\phi_\rho,*}$.
>
> The inequality in Theorem 4 immediately implies that $\lim_{n\to \infty} R^\epsilon(f_n)=R^\epsilon_*$, so $f_n$ is a minimizing sequence for the adversarial classification risk. Thus the $\rho$-margin loss is consistent.

---

> > ### Comment · Reviewer_QDX6 · 2023-08-14
> > **Response**
> >
> > Thanks for clarification. I misunderstood when reading the inequality in Theorem 4. Look forward to seeing the extension of Proposition 2.

---

### Official Review · Reviewer_6vgU · 2023-07-04

**Soundness:** 3 good
**Presentation:** 3 good
**Contribution:** 3 good
**Rating:** 7
**Confidence:** 3

**Summary:**

The paper provides a sufficient and necessary condition for surrogate loss to be adversarial consistency, and provide $\rho$-margin loss that satisfies the proposed condition so that it can replace 0-1 loss.

**Strengths:**

Understanding the consistency of loss in the adversarial setting is a rather ongoing topic and hasn’t been addressed yet. The paper studies the consistency of surrogate loss for robust binary classification and presents some examples of loss function that satisfies the calibration condition, which has a good contribution to the community. Past work shows no convex loss (which people often use in practice) is adversarially consistent. This paper further provides sufficient and necessary conditions regarding what kind of surrogate loss is adversarially consistent. The presentation is clear and easy to follow.

**Weaknesses:**

It shouldn’t be surprising that $\rho$-margin loss can be a good surrogate loss for adversarial training, as when $\rho$ is extremely small the loss is approximately 0-1 loss. Yet such loss is non-differentiable and therefore hard to use in practice.

**Questions:**

In the paper, the author also presents another shifted sigmoid loss that satisfies the consistency property. I wonder if the author can shed any light on the performance of such loss function used in adversarial training on some simple dataset like MNIST.

---

> ### Author Rebuttal · Authors · 2023-08-08
>
> - Somewhat surprisingly, there are surrogate losses very close to the $0$-$1$ loss that are not adversarially consistent. Let $\psi(\alpha)=1/(1+\exp(\alpha))$) be the sigmoid loss and define $\phi_K(\alpha)=\psi(K\alpha)$. Notice that for large $K$, $\phi_K$ is also a close approximation  to the 0-1 loss. However, our results imply that $\phi_K$ is not adversarially consistent.
>     This example illustrates that the consistency of the $\rho$ margin loss does not follow merely from the fact that it approximates the $0$-$1$ loss well.
>
> - Yes, running an experiment on MNIST comparing different losses would definitely be interesting. To limit the scope of this paper, we focus on theoretical aspects of adversarial learning and leave experiments for future work.

---

> > ### Comment · Reviewer_6vgU · 2023-08-10
> > **Replying to Rebuttal by Authors**
> >
> > Thank you for answering my questions. It's indeed surprising and a bit confusing, as the author mentioned in the paper that shifted sigmoid loss is adversarial consistent, but from the rebuttal, the scaled sigmoid loss without shifting isn't adversarial consistent. So seems for sigmoid loss the shifting is important. I'm curious whether the author has any intuitive explanation or it is purely based on the technical details in the proof.

---

> > > ### Author Response · Authors · 2023-08-11
> > >
> > > The fundamental reason losses satisfying $C_\phi^*(1/2)<\phi(0)$ are adversarial consistent is that minimizers of $C_\phi(\eta,\cdot)$ are bounded away from zero. Shifting the sigmoid function changes its behavior near zero, while scaling it does not. This difference accounts for their different properties.
> > >
> > > Notice that when  0 doesn't minimize $C_\phi(1/2, \cdot)$ the counterexample of proposition 2 breaks down, and so the condition $C_\phi^*(1/2)<\phi(0)$ rules out this counterexample.

---

> > > > ### Comment · Reviewer_6vgU · 2023-08-11
> > > >
> > > > I thank the author for answering my questions. I'll keep my score and vote for acceptance.

---

### Official Review · Reviewer_epdC · 2023-07-08

**Soundness:** 3 good
**Presentation:** 2 fair
**Contribution:** 4 excellent
**Rating:** 6
**Confidence:** 3

**Summary:**

The paper analyzes the consistency of surrogate losses in the case where there is an adversary that perturbs the sample data. Unlike in empirical risk minimization, where many convex surrogate losses have been proven to be consistent, the authors show that no convex surrogate losses are adversarially consistent. The authors provide a theoretical analysis to back up the claim. In addition, the authors design a non-convex surrogate loss that is adversarially consistent.

**Strengths:**

- The paper presents an important analysis of the consistency of surrogate losses under adversarial examples setting.
- The authors provide a thorough theoretical analysis of the property of surrogate losses under adversarial examples setting.
- The authors propose a surrogate loss that is adversarially consistent.

**Weaknesses:**

- The presentation of the paper is a bit hard to parse. For example, some of the notations are used before they are defined, like in the last few paragraphs of the introduction.
- To improve the clarity of the presentations, I suggest the authors incorporate some figures to illustrate the difference of consistency property in adversarial vs regular cases.
- It will be good to also show the benefit of the adversarially consistent surrogate loss empirically. It could be experiments on real-world data or even some synthetic data to demonstrate the effect of having adversarially consistent surrogate loss in practice

**Questions:**

Please answer my concerns in the previous section.


**Limitations:**

No concern.

---

> ### Author Rebuttal · Authors · 2023-08-08
>
> - We apologize we forgot to define $\tau$ towards the end of the introduction. In the shifted sigmoid loss, the constant $\tau$ is any positive number.
>     The remaining notation in this paragraph seems to be properly defined, but in our revision we will try to clarify this part of the paper further.
>
>
>  - Thank you for the suggestion to include figures. Space permitting, we will include several simple examples in the revised version to illustrate the main concepts.
>
>  - Yes, running an experiment on MNIST comparing different losses would definitely be interesting. To limit the scope of this paper, we focus on theoretical aspects of adversarial learning and leave experiments for future work.

---

> > ### Comment · Reviewer_epdC · 2023-08-13
> >
> > Thanks to the authors for the rebuttal.
> > I'd still strongly suggest including a simple experiment in the paper to show the benefit of consistent surrogate loss empirically, as a good theory should also be practical.

---

### Official Review · Reviewer_PJjd · 2023-07-10

**Soundness:** 3 good
**Presentation:** 2 fair
**Contribution:** 4 excellent
**Rating:** 6
**Confidence:** 3

**Summary:**

this paper tackles the problem of consistency in adversarial classification. Consistent losses are losses whose minimization lead to the minimization of the 0/1 loss. Although consistent losses are known for a long time in standard classification, they were not known in the adversarial setting. This paper show a simple necessary and sufficient condition for a loss to be adversarially consistent.

**Strengths:**

The existence of consistent losses in adversarial classification is a very important problem, and this paper solves it in some way.
Up to section 3, the paper is very well written and easy to understand.
Also, section 5 provides very useful bounds on margin losses, which are the counterpart of what was published in the non-adversarial setting by Bartlett, Zhang, and others.


**Weaknesses:**

section 4 is much harder to understand. For example, to understand propositions in that section, reading [Frank and Niles-Weed] helps a lot. In my opinion, authors should work on improving the clarity of this section.

details:
- Proposition 2, Line 139: a unit ball is of radius 1. But here, radius is R=epsilon/2
- Proposition 3, Line 211: recall that the inequality holds thanks to assumption 1, to make things clearer.
- Why put the proof of proposition 2 in the paper ? The proof is exactly the same as Meunier’s proof of the same result. Removing the proof would release space to improve clarity of section 4.


**Questions:**

1. It seems like thm1 is already known. In particular, points (1) and (2) boil down to the equivalence between calibration and consistency in the non adversarial setting, isn’t it right ?

2. In Proposition 4: aren’t \mathbb{P}*_0 and \mathbb{P}*_1 maximizers of \bar{R}_\phi instead of \bar{R} ? I don’t see the logic here.
Make explicit that there exists a pair \mathbb{P}*_0 and \mathbb{P}*_1 maximizing both \bar{R} and \bar{R}_\phi. I also don't see how it is related to the reference [Frank and Niles-Weed]. Please make this clearer.



**Limitations:**

limitations have been addressed.

---

> ### Author Rebuttal · Authors · 2023-08-08
>
>    Weaknesses:
>
>  - We will look for ways to clarify Section 4 by adding additional context for the main Propositions. Regarding  Proposition 2, the statement is of course crucial for our main theorem, and this example is both simple and illuminating.
>
>  -  Your comments on lines 139, 211: we have incorporated your feedback.
>
>
>
>    Questions:
>
>
>  1. Yes you are right. We included a proof of this result only because we couldn't find it stated in this precise form in prior work. We will clarify that proposition 1 is already known and add a citation to results on calibration.
>
>
>  2. Yes, that was a typo, thank you for finding this mistake! To clarify, $\mathbb P_0^*$ and $\mathbb P_1^*$ are maximizers of $\bar R_\phi$, and we do not need to assume that they are maximizers of $\bar R$.
>
>
>
>
>   Relation to the reference [Frank and Niles-Weed]:
>
>  Lemma 16 of [Frank and Niles-Weed] proves an approximate complementary slackness condition for a convex relaxation to $R_\phi^\epsilon$ which they call $\Theta$. Later, in Lemma 26, they prove that minimizing the convex relaxation $\Theta$ is equivalent to minimizing $R_\phi^\epsilon$. Combining the approximate complimentary slackness result (Lemma 16 of [Frank Niles-Weed]) together with the equivalence of minimizing $\Theta$ and $R_\phi^\epsilon$ (Lemma 26 of [Frank Niles-Weed]) results in Proposition 4 of our paper.
>   We will clarify this in our revised version.
>
>   To avoid confusion, we will include a self-contained proof of this result in our appendix.

---

> > ### Comment · Reviewer_PJjd · 2023-08-20
> >
> > Thank you for your answers. Yes, a self-contained proof would *definitely* improve the readability of the paper.

---

### Decision · Program_Chairs · 2023-09-21

**Decision:**

Accept (poster)

**Comment:**

## Summary

This paper studies consistency in adversarial classification. Specifically, they are concerned with losses that are good surrogates for the 0/1 loss (minimizing one leads to minimizing the other). In contrast to non-robust ML, such losses were not known for adversarial robustness before this paper. As one of the main results, the paper shows necessary and sufficient conditions for a loss function to be adversarially consistent. The reviewers found this to be novel, building on several prior works. One of the new contributions is that the authors' results apply to non-convex loss functions. One of the techniques in the paper translates the optimality of the adversarial loss into the optimal coupling and then uses a adversarial consistency analysis. The reviewers (after sufficient discussion) found the theoretical analysis to be correct and clear, and the reviewers also found the results overall to be compelling.

The reason the recommendation is not higher is simply that there was not enough support from the reviewers about the impact/originality/difficulty of the work. It seems like a solid contribution to the field.

## Improvements

There were no major concerns with the paper. However, there were several suggestions for improvements for the final version of the paper. For example:
- clarify that proposition 1 is already known
- fix several typos
- improve the introductions and add more definitions of notation
- add expository figures to illustrate the key ideas and concepts
- include some experiments to demonstrate the results (e.g., on synthetic toy examples)